# When RAD52 Allows Mitosis to Accept Unscheduled DNA Synthesis

**DOI:** 10.3390/cancers12010026

**Published:** 2019-12-19

**Authors:** Camille Franchet, Jean-Sébastien Hoffmann

**Affiliations:** Laboratoire d’Excellence Toulouse Cancer (TOUCAN), Laboratoire de pathologie, Institut Universitaire du Cancer-Toulouse, Oncopole, 1 avenue Irène-Joliot-Curie, 31059 Toulouse cedex, France; franchet.camille@iuct-oncopole.fr

**Keywords:** DNA replication, replication stress, mitotic DNA synthesis, RAD52, chromosome instability, genome instability

## Abstract

Faithful duplication of the human genome during the S phase of cell cycle and accurate segregation of sister chromatids in mitosis are essential for the maintenance of chromosome stability from one generation of cells to the next. Cells that are copying their DNA in preparation for division can suffer from ‘replication stress’ (RS) due to various external or endogenous impediments that slow or stall replication forks. RS is a major cause of pathologies including cancer, premature ageing and other disorders associated with genomic instability. It particularly affects genomic loci where progression of replication forks is intrinsically slow or problematic, such as common fragile site (CFS), telomeres, and repetitive sequences. Although the eukaryotic cell cycle is conventionally thought of as several separate steps, each of which must be completed before the next one is initiated, it is now accepted that incompletely replicated chromosomal domains generated in S phase upon RS at these genomic loci can result in late DNA synthesis in G2/M. In 2013, during investigations into the mechanism by which the specialized DNA polymerase eta (Pol η) contributes to the replication and stability of CFS, we unveiled that indeed some DNA synthesis was still occurring in early mitosis at these loci. This surprising observation of mitotic DNA synthesis that differs fundamentally from canonical semi-conservative DNA replication in S-phase has been then confirmed, called “MiDAS”and believed to counteract potentially lethal chromosome mis-segregation and non-disjunction. While other contributions in this Special Issue of *Cancers* focus on the role of RAS52RAD52 during MiDAS, this review emphases on the discovery of MiDAS and its molecular effectors.

## 1. The Conventional DNA Replication Program and the Responses to Replicative Stress

The duplication of chromosomes during S phase of the cell cycle in multicellular organisms contributes vastly to cell survival and evolution by ensuring the maintenance of genome integrity and the required adaptive responses to endogenous or external genotoxic stresses. The DNA replication process starts shortly after mitosis, during the G1 phase of the cell cycle, when daughter cells organize their genomes into large DNA replication domains containing multiple initiation sites that will be activated simultaneously in S phase. From these replication origins progress thereplication forks which ensure stable genetic and epigenetic inheritance. In human cells, the process takes about 10 h and involves the activation of roughly 50,000 replication origins [1]. The accurate elongation of these forks on undamaged genomic DNA requires the action of the most abundant replicative DNA polymerases δ and ε which perform the duplication of the six billion nucleotides that constitute the human genome [2]. However, nature needs more flexibility and when the replication complex encounters endogenous DNA distortions within repetitive sequences as well as non-B DNA structures [3,4] or persistent base modifications by exogenous aggressions such as chemical carcinogens and ionizing radiation, it frequently stands. This is due to the high selectivity of these replicative DNA polymerases which are unable to accurately insert a base opposite a damaged base or a base engaged in structural DNA perturbations, a phenomenon referred as replicative stress (RS) that strongly affects genome stability. Natural replication barriers include also compacted chromatin, protein–DNA complexes as well as conflicts between replication forks and transcription, a type of collision incident of intense interest [5] that can generate important torsional stress leading to replication fork reversal. RS is an important feature during oncogene-driven cancer progression and is a major source of the unstable cancer genomes [6,7]. Indeed, failure to stabilize and restart stalled forks or prolonged arrest of replication forks may result in fork collapse, leading to chromosomal breakage and rearrangement. Besides the problem of fork progression itself, RS can also be explained by some oncogene-driven mechanisms based on usage of replication origins, which could be insufficient or excessive [8] resulting all in replication fork breakage. Overexpression of the cyclin E oncogene can affect the binding onto chromatin in G1 of the MCM helicases, important component of the pre-replication complexes (pre-RCs), resulting in a rarity of pre-RCs to allow completion of S phase [9]. Conversely, excessive origin firing induced by overexpression of RAS and MYC oncogenes results in severe depletion of the cellular pools of dNTPs and ultimately triggers replication fork stalling [10].

To avoid an aberrant interruption of the cell cycle caused by the impediment of DNA replication, human cells have evolved multiple options to deal with the constant challenge of RS, depending on the source of the stress, the nature of the blockage and the level of accumulated stalled forks. Since stalled forks are frequently associated with large amounts of unwound single-stranded DNA (ssDNA) covered by the protein RPA, it is believed that the major signal for many responses to RS is the generation of this RPA-coated ssDNA. This is the case for the activation of the replication checkpoint, the main response that senses stalled forks in S phase, activates its cardinal kinase ATR, that in turn phosphorylates hundreds of substrates in order to stabilize and restart the stalled DNA forks [11]. Compensation by the activation of new replication origins, called “dormant origins, in the vicinity of stalled forks upon RS is another option that the cells use to respond to RS [12]. Indeed, the MCMs are loaded onto DNA in a large excess when compared to the number of active replication origins and such excess of MCM is thought to provide a reservoir of dormant origins that can be activated if needed. In normal growth conditions, dormant origins do not fire and are passively replicated by the fork coming from adjacent activated origins, while upon RS they are waked up so the inhibition of replication can be compensated, and the stalled forks rescued. In the case of blocking lesions, the arrested replicative DNA polymerase needs to be transiently substituted by a translesion-synthesis (TLS) DNA polymerase that is competent for the lesion bypass [13]. Because the TLS DNA polymerases do not have a proofreading function, TLS is inevitably mutagenic.

If active replication origins are lacking in the neighborhood of a stalled fork or if the responses to RS described above fail, this favors fork collapse and DNA double-strand break (DSB) are created at the stalled replication forks [14], a phenomenon known as replication fork breakage. In response to extensive RS and after prolonged stalling of the forks, the endonuclease MUS81 which forms a complex with EME1 or EME2 is responsible for such breakage in mitosis [15]. Recently, we unveiled that the endonucleases ARTEMIS and XPF-ERCC1 can also induce stalled DNA replication forks cleavage in G2 [16]. Such a mechanism of fork rescue that involves DSB occurring in late stages of the cell cycle (G2 or M) could represent one of the last chances to rescue stalled forks in order to prevent mitotic segregation defects. Indeed, these DSB can be resolved by repair mechanisms such as the highly conservative homologous recombination (HR) or the non-homologous end-joining (NHEJ).

## 2. The Discovery of MiDAS

The failure of the RS response also increases the opportunity of incompletely replicated loci or unresolved replication intermediates and a fraction of these under-replicated genomic loci have been evidenced to enter into mitosis. When not correctly handled in the M phase, such DNA regions are converted into complex broken-DNA structures that are transmitted to G1 daughter cells where they are sequestered and shielded in nuclear compartments described as nuclear 53BP1 bodies [17,18]. To limit this dangerous transmission, the cells possess a very last chance mechanism during cell cycle to replicate these under-replicated loci, the mitotic DNA synthesis or MiDAS. We reported the first observation of this unexpected extremely late DNA synthesis in 2013, during investigations aiming at exploring the mechanism by which the specialized DNA polymerase eta (Pol η) contributes to the replication and stability of chromosomal common fragile sites (CFS) [19]. CFS are large chromosomal regions that are usually replicated late in S phase and are frequently rearranged in tumors [20]. Their instability, which occurs in early stages of cancer development, is believed to result from the combination of challenges to fork movement by non-B DNA secondary structures and a rarity of replication initiation events able to compensate the stalled forks [21]. We presented evidences that human Pol η, a TLS DNA polymerase best known for its role in responding to UV irradiation-induced genome damage, can be recruited to some CFS during S phase in normal growth condition or upon a mild replicative stress for the replication of non-B structured DNA within these CFS. We also demonstrated that deficiency of Pol η resulted in the persistence of under-replicated at several CFS in mitotic cells and discovered a strong delayed replication of these regions. Indeed, by using an original in situ incorporation assay with the thymidine analogue EdU, we could detect EdU spots in Pol η -depleted G2 and/or mitotic cells as well as in G2 and/or mitotic cells from an XP-V patient [19] (see Figure 1A). Since the frequency of these EdU spots increased following aphidicolin treatment, we thought that this late DNA synthesis process may occur more generally upon replication inhibition. Later on, the group of Ian Hickson further documented a late replication completion at CFS upon aphidicolin treatment and demonstrated that it happened in early mitosis [22]. These authors reasoned that any EdU found in mitotic cells following an EdU pulse of 20 to 30 min was expected to have been incorporated in G2. Therefore, they used a CDK1 inhibitor to arrest cells at the G2/M transition which allowed them to separate G2 from Mitotic cells and add EdU only after cells had entered the prophase. Under these conditions, they observed EdU on metaphase chromosomes, demonstrating that DNA synthesis was occurring in early mitosis. They also showed that this mitotic DNA synthesis, a process they called MiDAS, corresponded to broken CFS loci by combining EdU detection with fluorescent in situ hybridization (FISH) [22]. Collectively, these recent observations demonstrate that, while S-phase was traditionally believed to end before the G2-phase of the cell cycle, it can extend almost to the stage of cell division, at least at some intrinsically hard-to-replicate chromosomal domains such CFS.

## 3. Why the Cells Use MiDAS?

To date, the loci that most depend upon MiDAS seem to be the difficult-to-replicate DNA sequences such CFS and telomeres, very sensitive to even mild replicative stress [25]. More generally, such exceedingly late replicating regions of the genome might correspond to zones where two converging replication forks have both stalled in a region that lacks dormant origins and a successful replication completion of these loci by MiDAS could limit the formation of ultrafine anaphase bridges (UFBs) which compromise the partition and integrity of chromosomes [26]. MiDAS could also limit the transmission of DNA damage to the next generation of cells where a new chance will be offered to complete a faithful replication at these loci [27].

MiDAS has been documented to occur in several different stress contexts. We recently reported a two-fold increase in the percentage of cells with MiDAS in BRCA2-deficient compared to BRCA2-proficient mitotic cells (Figure 1B). We showed that removing MUS81 in BRCA2 deficient cells reduced the percentage of EdU-positive mitotic chromosomes, suggesting that cells lacking BRCA2 completes DNA replication during mitosis and implicated MUS81 in this process [23]. We have also evidenced that replicative stress stimulates MiDAS at telomeric sequence in the absence of Pol η in cells that maintain their telomere through the alternative lengthening of telomeres (ALT) mechanism [24] (Figure 1C). About 10% of all cancers, including some that have a very poor prognosis, use this ALT pathway to prevent telomere shortening [28]. ALT-positive cells usually have a number of characteristics, including telomeric DNA that is separated from chromosome ends and forms partially single-stranded circles. Telomere elongation in ALT cells involves homologous recombination actors such the MRN complex (MRE11/RAD50/NBS1), SMC5/SMC6, FEN1, MUS81, FANCD2 and FANCA, able to restart stalled telomeric DNA replication [28]. We found that Pol η localized specifically to ALT telomeres and regulated the ALT mechanism. When we treated Pol η-depleted ALT cells with low dose of aphidicolin followed by EdU pulse, EdU positive newly synthesized DNA was observed on telomeric sequences of metaphase chromosomes, demonstrating that MiDAS is also a feature of human telomeres [24] (Figure 1C). In fact, a recent work reported that telomeric MiDAS seems also to occur in cells maintaining their telomeres by the telomerase [29].

Collectively, these findings imply that any form of replicative stress, including DNA insults from external or internal sources but also genetic events that slow replication fork progression can all lead to MiDAS at hard-to-replicate DNA sequences.

## 4. Mechanisms and Molecular Actors Involved in MiDAS

More than a late DNA replication mechanism, MiDAS could be better considered as a DNA repair process when conventional DNA replication forks are converted during mitosis into unprotected genomic DNA that can be targeted by repair factors. The attempted condensation of incompletely duplicated loci in early mitosis has been proposed to be the cause of this late DNA synthesis. It has been recently reported that the switch from conventional DNA replication to MiDAS could be facilitated by the TRAIP ubiquitin ligase that triggers replisome disassembly in response to unfinished DNA replication, providing access for nucleases and the MiDAS repair factors at unprocessed parental DNA [30]. Upon mitotic entry, CDK1-dependent phosphorylation of EME1 promotes the formation of the complex with the nuclease MUS81 and SLX4 and their recruitment to under-replicated CFS [22,31,32] a step required for the subsequent MiDAS. The catalytic and non-catalytic subunits of the replicative DNA polymerase δ, POLD1 and POLD3 respectively, are both associated with mitotic chromatin after replicative stress but POLD3 depletion only prevents MiDAS and generates increased levels of DNA strand breaks on metaphase chromosomes, suggesting that MiDAS at metaphase chromosome gaps corresponds to de novo DNA synthesis promoted by the POLD3-associated DNA polymerase δ complex. Collectively, these observations lead to a model where stalled forks in early mitosis could be cleaved by the SLX4/MUS81/EME1 complex triggering a POLD3-dependent DNA synthesis at CFS (Figure 2). Since concomitant depletion of MUS81 and POLD3 is not enough to fully eradicate MiDAS [23], an alternative MUS81- or POLD3-independent mechanism might contribute to its activation. 

In yeast, it is well established that Rad52 is a recombination factor that allows the exchange of RPA with Rad51 on ssDNA thus promoting homologous recombination (HR) [33]. In human cells, this exchange activity is carried out by BRCA2 and the function of Rad52 in DNA replication and repair is less well defined [34]. RAD52 consists of an N-terminal ssDNA annealing domain, central replication protein A (RPA) interacting domain and C-terminal RAD51 mediating domain [35]. While it is habitually believed that human RAD52 acts in the rescue of perturbed replication forks under pathological conditions, recent work support that it contributes to the integrity of stalled replication forks in normal cells by stabilizing the nascent strand after fork arrest and preventing an excessive action of fork reversal enzymes [36]. RAD52 is recruited to nuclear foci after replication fork arrest and acts at the exposed parental ssDNA formed once the fork gets blocked. Its ability to interact with ssDNA and RPA and its aptitude to multimerize make this factor beneficial in single-stranded DNA annealing into regions of micro-homology (Figure 2) and preventing excessive loading of fork reversal enzymes such SMARCAL1 and subsequent extensive nascent strand degradation by MRE11 [36]. Therefore, the role of RAD52 is essential for a correct recovery from the replication arrest in human cells and RAD52 inhibition results in under-replication and a strong dependence on RAD51 and BRCA2 for viability [35,37]. 

The contribution of RAD52 to DNA repair has also been described recently. When a sister chromatid is present as a donor, RAD51-mediated gene conversion generally occurs after passage of the replication fork while under-replicated DNA is located within the template ahead of the incoming fork and thus requires replication-coupled repair named break-induced replication (BIR), a form of conservative DNA synthesis that occurs when a broken chromosome invades its homologous duplex, initiates at broken or processed replication forks and proceeds along the length of the chromosome. RAD52, by promoting single-stranded DNA capture and through the creation of a higher-order structural form of RAD52, seems to hold an important role of DNA annealing in this process [38]. It is loaded onto mitotic chromatin following replicative stress and RAD52 foci co-localized with FANCD2 twin foci in early mitosis together with RPA and MUS81, whereas RAD51 and BRCA2 do not and the RAD52-mediated BIR was shown to drive MiDAS at the final stages of the cell cycle [39]. Indeed, while depletion of either BRCA2 or RAD51 increases the number of MiDAS foci (FANCD2-associated EdU foci in prometaphase cells), RAD52 depletion or inhibition clearly abolishes MiDAS and increases mitotic chromosome missegregation [39]. Importantly, it was reported that in mitosis the recruitment of RAD52 to chromatin depends on SLX4 and depletion of RAD52 affects the loading of MUS81, thus impairing MiDAS and therefore increases the generation of UFBs, chromatin bridges and micronuclei. Therefore, RAD52 appears to hold a cardinal function at an SLX4-dependent step in CFS-associated MiDAS prior to the involvement of MUS81 or POLD3 (see model in Figure 2). At telomeres, MiDAS resembles to CFS-related MiDAS as it requires also RAD52 [29] but it does not require the MUS81-EME1 endonuclease, suggesting that another nuclease might be involved or MiDAS events do not all require an initial endonuclease cut.

## 5. Targeting RAD52 as a Novel Anti-Cancer Treatment Strategy

Because RS is a highly relevant mechanism to explain genomic instability in cancer cells, its studies are at the forefront of cancer research today. Indeed, the persistence of stalled forks can lead to fork collapse and under-replicated regions that in turn leads to chromosomal breakage and chromosome instability (CIN). If we consider genomic instability and the resulting unstable cancer genome from an evolutionary perspective, CIN provides a continuing pool of variants upon which selection could act, and therefore constitutes a driving force of tumor heterogeneity and development. Similar to the Darwin’s evolutionary tree of speciation, clonal evolution in cancer can be explained by the multiple forms of selective pressure including physical barriers, environment and nutrients, oxygen concentration, immunity, and therapy, allowing some mutant sub clones to expand while others go extinct [40]. CIN promoting tumor heterogeneity and drug resistance, it is generally associated with poor prognosis. However, extremes of CIN correlate with enhanced cancer outcome [41], supporting the notion of an appropriate threshold of RS and genetic instability for tumor viability [42], selection ensuring a necessary karyotypic diversity to survive upon environment changes but limiting RS/CIN to prevent excessive genome instability, deleterious for cell fitness.

Several years ago, we made the hypothesis that expression of specific genes that could neutralize excess RS in cancer cells might correlate with poor patient survival and we therefore explored the prognosis value of the expression of more than 100 genes involved in replication origins, fork stabilization, fork progression, replication checkpoint and repair of collapsed forks in large cohorts of solid cancer and hematological malignancies. Thus far, we found two categories of biomarkers as strong predictors of patient survival and both of them are indeed identified to limit excessive RS: the first category regroups the genes coding the checkpoint mediators Chk1, Claspin and Timeless known to stabilize the stalled replication forks upon RS [43,44]. The second category includes the genes POLQ and POLN encoding the A-family DNA polymerases Pol θ and Pol ν [45,46,47,48,49]. In these studies, the strongest marker associated with patient survival was POL Q, the gene encoding Pol θ, which holds a cardinal role in an alternative form of end-joining, referred to as microhomology-mediated end-joining (MMEJ) that substitutes for HR deficiency for repairing the breaks associated with RS [50,51]. Collectively, these studies suggest that Chk1, Claspin, Timeless and Pol θ by counteracting karyotypic diversity, could contribute to tumor progression and constitute important novel targets in cancer therapy (Figure 3). The molecular actors of MiDAS were not tested in these studies, but it is conceivable that MiDAS might also represent typically one of the options used by cancer cells to counteract constitutively high levels of RS by limiting the persistence of under-replicated DNA and avoiding excessive karyotypic diversity (Figure 3). Therefore, we may exploit therapeutically the MiDAS protective role of RAD52 for tumors with excessive RS, especially those exposed to genotoxic agents already used in therapy that generate additional RS. Manipulating RS in tumors, as a targeted therapeutic opportunity, currently represents a very active area of research and there is an increasing clinical interest in the use of small molecule inhibitors that exacerbate RS in human cancers (inhibitors of PARP, Chk1, Pol θ). RAD52 inhibitors need to be added to the list, especially because similarly to Polθ, RAD52 inhibition might be particularly selective, given its limited role in genome protection in normal cells [52,53]. Interestingly, RAD52 inactivation leads to increased cell death in lung tumors and in BRCA2-deficient cancer cells [35,37]. It is tempting to speculate that this could be linked to the abrogation of MiDAS.

The concept of synthetic lethality by PARP inhibitors has been successfully used in therapy since 2015 for ovarian cancer. PARP inhibitors are now considered as a standard of care in patients with ovarian, breast and prostate cancers (this will be the case shortly for patients with pancreatic cancer) in the context of HR-deficiency (germinal or somatic BRCA1/2 inactivation). Despite interesting results, there is a high rate of primary or secondary resistance to PARP inhibitors. Several mechanisms involved in resistance might be linked to the responses to RS and targeting MiDAS, might be an interesting option to test. Interestingly, synthetic lethality can also be achieved by RAD52 depletion in HR-deficient human cancer cells [54]. This synthetic lethality depends on the endonuclease EEPD1, which cleave stalled replication fork and produced a toxic intermediate. Therefore, RAD52 blockade in patients with HR-deficient tumors has become one of the most promising therapeutic option in the field of RS-associated strategies for cancer treatment [55]. Several small molecules have been described. One of the most studied is 6-hydroxyDL-dopa [56]. 6-hydroxyDL-dopa inhibits RAD52 binding to ssDNA, leading to improved cell death in BRCA1 deficient triple negative breast cancer cell line MDA-MB436 and in FLT3 acute myeloid leukemia cells, alone or in association with PARP inhibitors [56,57]. Other molecules inhibit ssDNA/RAD52 interaction such as AICAR and (AICAR) 5′ phosphate (ZMP) with significant effect on BRCA1-mutated HCC1937 cells and BRCA2-mutated Capan-1 cells [58]. Other inhibitors act by inhibiting the ssDNA annealing activity of RAD52, such as D-103 and D-G23 [59]. Although many molecules are simultaneously developed, giving promising results both in vitro and in vivo, RAD52 and MiDAS remain poorly monitored in clinical trials and none of the RAD52 inhibitors cited above are actually in clinical stages of development.

## 6. Conclusions

In this review, we have analyzed the current views on how late unscheduled DNA synthesis, orchestrated by RAD52, can represent the last opportunity during one cell cycle to respond to replication stress and limit chromosome instability and transmission of DNA damage to the next cell generation. Whether MiDAS occurs through a break-induced replication (BIR)-like process needs further investigation. Besides stabilization of stalled DNA replication forks and repair process of broken forks, MiDAS is a critical response to replicative stress that should be considered as an important target in cancer therapy. The treatment of patients with MiDAS inhibitors, alone or in combination with other agents limiting excessive RS, might selectively target tumors exhibiting RS induced by oncogene activation.

## Figures and Tables

**Figure 1 cancers-12-00026-f001:**
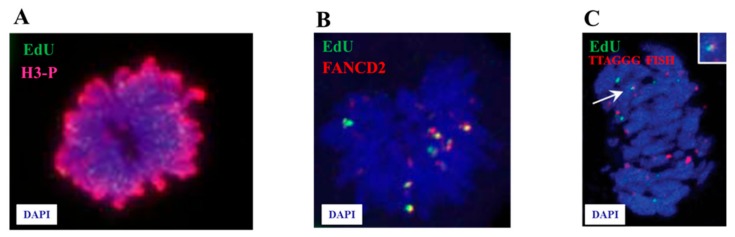
Illustrations of MiDAS at CFSs and telomeres in human cells. (**A**) From [19]: original and first published experiment showing EdU incorporation in mitotic cells. Asynchronously growing cells treated with 0.3 µM Aphidicolin for 24 h were labeled with EdU for 45 min, fixed and stained with a phospho-H3 (Ser10) specific antibody. DNA was counterstained with 4′,6-Diamidine-2′-phenylindole dihydrochloride (DAPI). The presence of EdU signals on p-H3-positive mitotic chromosomes can be observed. (**B**) From [23]: H1299 cells expressing a BRCA2 shRNA were pulsed with EdU for 40 min and processed for detection of EdU (green) and immunofluorescence staining with anti-FANCD2 antibody (red) on metaphase spreads. DNA was counterstained with DAPI. (**C**) From [24]: Pol η depletion leads to MiDAS at telomeres in U2OS cells treated for 24 hr with aphidicolin and then for 40 min with EdU. EdU (green) and Fluorescence in situ hybridization with the telomeric probe TTAGGG (red) were detected on metaphase spreads. DNA was counterstained with DAPI. Scale bar, 10 μm.

**Figure 2 cancers-12-00026-f002:**
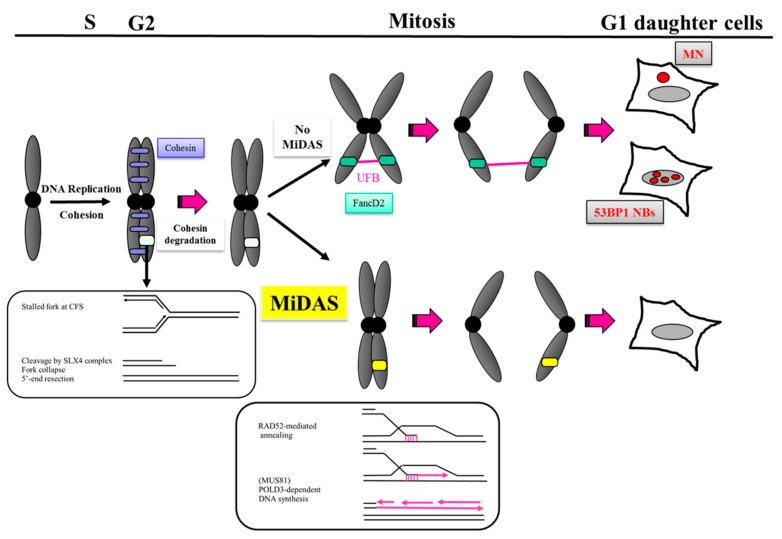
Model for MiDAS function and mechanism. Successful replication completion of CFS loci by MiDAS could limit the formation of ultrafine anaphase bridges (UFBs) which compromise the partition and integrity of chromosomes and reduce the transmission of DNA damage (53BP1 NBs) to the next generation of cells. One model proposed that MiDASoccurrs through a break-induced replication (BIR)-like process after cohesin degradation. Persisting stalled replication forks at difficult-to-replicate region within CFS upon mild replicative stress could result in under-replicated DNA (illustrated as a white locus on chromosomal arm) and trigger its collapse by the action of SLX4 in complex with the MUS81-EME1 endonuclease in early mitosis. The vicinity of the sister chromatid at the collapsed fork renders unnecessary an extensive homology search and a HR process to start while a limited end resection favors the single-stranded DNA annealing activity of RAD52 into regions of micro-homology and the subsequent POLD3-dependent conservative DNA repair synthesis (EdU incorporation in a single chromatid).

**Figure 3 cancers-12-00026-f003:**
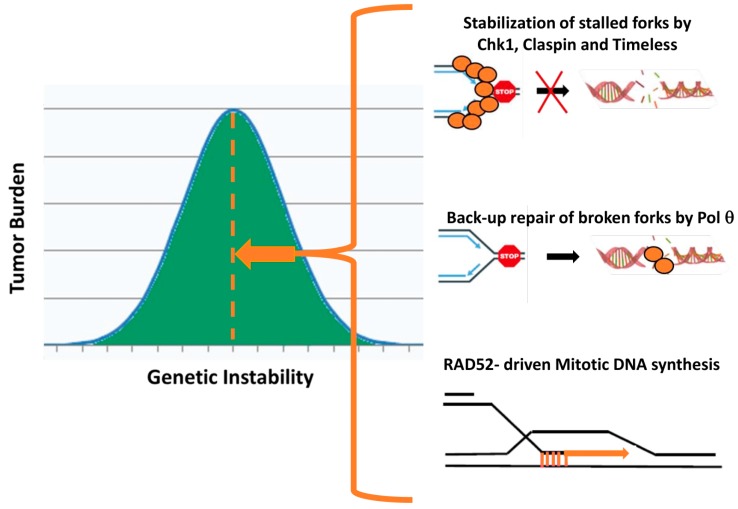
Three major responses to excessive RS that represent potential targets for cancer therapy. If genetic instability is important for tumor development, conversely excessive chromosome instability appears to suppress tumorigenesis and extreme genetic instability correlates with improved cancer outcome, suggesting that karyotypic diversity required to adapt to selection pressures might be balanced in tumors against the risk of excessive instability. We propose that three major mechanism known to respond to replicative stress could counteract karyotypic diversity, contribute to tumor progression and constitute important novel targets in cancer therapy: (i) stabilization of stalled forks by high expression of Chk1, Claspin and Timeless; (ii) overexpression of Pol θ that functions in a back-up alternative repair pathway at resected double-strand breaks when homologous recombination is deficient to repair broken forks; (iii) MiDAS mediated by RAD52.

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
