# Peer review of "When RAD52 Allows Mitosis to Accept Unscheduled DNA Synthesis"

_cancers, 2019, doi:10.3390/cancers12010026_

Round 1
Reviewer 1 Report
The article provides a comprehensive overview of the discovery, the functions, as well as the molecular apparatus of a unique replication program executed during Mitosis termed MiDAS. Special emphasis is given to the role of RAD52 in this process. This aspect is subsequently further used to develop a rational for cancer treatment strategies targeting RAD52 and other factors implicated in the process of MiDAS in order to selectively kill cancer cells.
The article is well-written and provides useful background on the process of MiDAS and its origins. It was a pleasure to read.
Suggestions for improving the presentation follow:
Minor comments:
Summary, line 28: RAS52 should be RAD52. In some instances along the text the authors tend to use long sentences, which unnecessarily complicate the message to be conveyed. It is advisable to rephrase such sentences. Examples: page 2, line 44-51; page 5, line 203-208. There are more …. Figure 1 proposes a model implicating break induced replication (BIR) in the initiation of MiDAS. However, the schematic is rudimentary and not fully in line with the associated text. The paper will benefit from an improved Figure with more details on the proposed mechanism. It is stated by the authors that MiDAS can be documented in mitosis using multiple techniques. As the authors are involved in discovery and characterization of this effect, it will help the uninitiated reader to include representative images with mitotic cells positive for MiDAS.
Author Response
We thank the reviewer for sending us insightful reviews.
As required, we have shortened some long sentences along the manuscript, we have included an additional figure illustrating some published images of MiDAS (Figure 1) and we provide the readers with a more detailed figure on the mechansim and the role of MiDAS (Figure 2).
We hope that the reviewer now find the manuscript to be suitable for publication in 'Cancers"
Reviewer 2 Report
This review provides an overview of our current knowledge regarding the recently discovered process mitotic DNA synthesis (MiDAS), and its role as a last chance response by cells to replication stress.
Overall this review was well written and informative, particularly during the sections that provide an overview of the replicative stress response, background on MiDAS and the major players involved, as well as what is currently known about the involvement of RAD52 in this process.
The area that could use a slight revision is section 5, which discusses RAD52 as an anti-cancer treatment strategy. This section could use some more recent citations regarding the topic of RAD52 inhibitors. There are a few more recently published papers that focus on the development of RAD52 small molecule inhibitors as an anti-cancer therapy, particularly against cancers that are deficient in other DNA repair mechanisms.
Author Response
We thank the reviewer who found that our review is well written and informative. We totally agree that we should better describe recently published papers that focus on the development of RAD52 small molecule inhibitors as an anti-cancer therapy. We have now included in our revised manuscript a new paragraph describing RAD52 inhibitors.
We hope that the reviewer now find the manuscript to be suitable for publication in 'Cancers"
Reviewer 3 Report
In this mini-review, Franchet and Hoffmann summarized the current research progress of mitotic DNA synthesis (MiDAS). The authors focused on the molecular mechanisms involved in MiDAS, particularly RAD52. The review is logically structured and well written, and the contents reflected the authors' expertise in MiDAS field. I only have a few questions.
The authors propose that 3 major mechanisms to counteract to replicative stress could counteract karyotypic diversity. Is there a synthetic lethal relationship between any of the two? Given the role of RAD52 in MiDAS, it will be interesting to discuss/speculate whether tumors with certain genomic backgrounds are hypersensitive to RAD52 inhibition/depletion. One example is that HR-deficient cancers rely on POLQ for survival. the quality of figures can generally be improved. they are currently over-simplified. If the authors can include more information in the figures, it will be very helpful for the readers
Author Response
We thank the reviewer who found our review logically structured and well written and for raising an interesting issue on the synthetic lethal relationship between any of the mechanisms that we descibe here to counteract karyotypic diversity. As pointed out by the reviewer, it is true that HR-deficient cancers rely on POLQ for survival. We noticed at the end of our manuscript that this is also true for RAD52: page 7, lanes 300-302: "Interestingly, RAD52 inactivation leads to increased cell death in lung tumors and in BRCA2-deficient cancer cells (Feng, Scott et al. 2011, Lok, Carley et al. 2013)". We also have included a novel paragraph on existing RAD52 inhibitors and synthetic letalityn with RAD52 knocking out.
As required, we have also improved the quality of figures : we have added Figure 1 illustrating some examples of MiDAS images and we provide now the readers with a more detailed figure on the function and mechanism of MiDAS in the new figure 2.
We hope that the reviewer now find the manuscript to be suitable for publication in 'Cancers"
Round 2
Reviewer 2 Report
The Authors responded to my comments.